# Amortized Causal Discovery: Learning to Infer Causal Graphs from Time-Series Data

## Abstract

Standard causal discovery methods must fit a new model whenever they encounter samples from a new underlying causal graph. However, these samples often share relevant information – for instance, the dynamics describing the effects of causal relations – which is lost when following this approach. We propose Amortized Causal Discovery, a novel framework that leverages such shared dynamics to learn to infer causal relations from time-series data. This enables us to train a single, amortized model that infers causal relations across samples with different underlying causal graphs, and thus makes use of the information that is shared. We demonstrate experimentally that this approach, implemented as a variational model, leads to significant improvements in causal discovery performance, and show how it can be extended to perform well under hidden confounding.

## 1 Introduction

Inferring causal relations in observational time-series is central to many fields of scientific inquiry (Berzuini et al., 2012; Spirtes et al., 2000). Suppose you want to analyze fMRI data, which measures the activity of different brain regions over time — how can you infer the (causal) influence of one brain region on another? This question is addressed by the field of *causal discovery* (Glymour et al., 2019). Methods within this field allow us to infer causal relations from observational data - when interventions (e.g. randomized trials) are infeasible, unethical or too expensive.

In time-series, the assumption that causes temporally precede their effects enables us to discover causal relations in observational data (Peters et al., 2017); with approaches relying on conditional independence tests (Entner and Hoyer, 2010), scoring functions (Chickering, 2002), or deep learning (Tank et al., 2018). All of these methods assume that samples share a single underlying causal graph and refit a new model whenever this assumption does not hold. However, samples with different underlying causal graphs may share relevant information such as the dynamics describing the effects of causal relations. fMRI test subjects may have varying brain connectivity but the same underlying neurochemistry; social networks may have differing structure but comparable interpersonal relationships; different stocks may relate differently to one another but obey similar market forces. Despite a range of relevant applications, inferring causal relations across samples with different underlying causal graphs is as of yet largely unexplored.

In this paper, we propose a novel causal discovery framework for time-series that embraces this aspect: Amortized Causal Discovery (Fig. 1). In this framework, we learn to infer causal relations across samples with different underlying causal graphs but shared dynamics. We achieve this by separating the causal relation prediction from the modeling of their dynamics: an amortized encoder predicts the edges in the causal graph, and a decoder models the dynamics of the system under the predicted causal relations. This setup allows us to pool statistical strength across samples and to achieve significant improvements in performance with additional training data. It also allows us to infer causal relations in previously unseen samples without refitting our model. Additionally, we show that Amortized Causal Discovery allows us to improve robustness under hidden confounding by modeling the unobserved variables with the amortized encoder. Our contributions are as follows:

- We formalize Amortized Causal Discovery (ACD), a novel framework for causal discovery in time-series, in which we learn to infer causal relations from samples with different underlying causal graphs but shared dynamics.

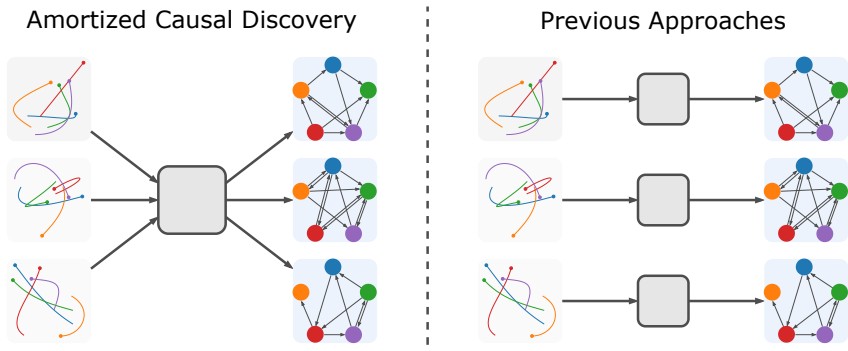

Figure 1: Amortized Causal Discovery. We propose to train a single model that infers causal relations across samples with different underlying causal graphs but shared dynamics. This allows us to generalize across samples and to improve our performance with additional training data. In contrast, previous approaches (Section 2) fit a new model for every sample with a different underlying causal graph.

- We propose a variational model for ACD, applicable to multi-variate, non-linear data.
- We present experiments demonstrating the effectiveness of this model on a range of causal discovery datasets, both in the fully observed setting and under hidden confounding.

## 2 BACKGROUND: GRANGER CAUSALITY

Granger causality (Granger, 1969) is one of the most commonly used approaches to infer causal relations from observational time-series data. Its central assumption is that causes precede their effects: if the prediction of the future of time-series $Y$ can be improved by knowing past elements of time-series $X$, then $X$ "Granger causes" $Y$. Originally, Granger causality was defined for linear relations; we follow the more recent definition of Tank et al. (2018) for non-linear Granger causality:

**Definition 2.1.** *Non-Linear Granger Causality*: Given $N$ stationary time-series $\boldsymbol{x} = \{\boldsymbol{x}_1, ... \boldsymbol{x}_N\}$ across time-steps $t = \{1, ..., T\}$ and a non-linear autoregressive function $g_j$, such that

$$\boldsymbol{x}_j^{t+1} = g_j(\boldsymbol{x}_1^{\leq t}, ..., \boldsymbol{x}_N^{\leq t}) + \boldsymbol{\varepsilon}_j^t \quad , \tag{1}$$

where $\boldsymbol{x}_j^{\leq t} = (..., \boldsymbol{x}_j^{t-1}, \boldsymbol{x}_j^t)$ denotes the present and past of series $j$ and $\boldsymbol{\varepsilon}_j^t$ represents independent noise. In this setup, time-series $i$ Granger causes $j$, if $g_j$ is not invariant to $\boldsymbol{x}_i^{\leq t}$, i.e. if $\exists\, \boldsymbol{x}_i'^{\leq t} \neq \boldsymbol{x}_i^{\leq t} : \ g_j(\boldsymbol{x}_1^{\leq t}, ..., \boldsymbol{x}_i'^{\leq t}, ..., \boldsymbol{x}_N^{\leq t}) \neq g_j(\boldsymbol{x}_1^{\leq t}, ..., \boldsymbol{x}_i^{\leq t}, ... \boldsymbol{x}_N^{\leq t})$.

Granger causal relations are equivalent to causal relations in the underlying directed acyclic graph if all relevant variables are observed and no instantaneous[1] connections exist (Peters et al., 2013; 2017, Theorem 10.1).

Many methods for Granger causal discovery, including vector autoregressive (Hyvärinen et al., 2010) and more recent deep learning-based approaches (Khanna and Tan, 2020; Tank et al., 2018; Wu et al., 2020), can be encapsulated by a particular framework:

1. Define a function $f_\theta$ (an MLP in Tank et al. (2018), a linear model in Hyvärinen et al. (2010)), which learns to predict the next time-step of the test sequence $\boldsymbol{x}$.

2. Fit $f_\theta$ to $\boldsymbol{x}$ by minimizing some loss $\mathcal{L}$: $\theta_\star = \mathrm{argmin}_\theta\, \mathcal{L}(\boldsymbol{x}, f_\theta)$.

3. Apply some fixed function $h$ (e.g. thresholding) to the learned parameters to produce the Granger causal graph estimate for $\boldsymbol{x}$: $\hat{\mathcal{G}}_{\boldsymbol{x}} = h(\theta_\star)$. For instance, Tank et al. (2018) infer the Granger causal relations through examination of the weights $\theta_\star$: if all outgoing weights $\boldsymbol{w}_{ij}$ between time-series $i$ and $j$ are zero, then $i$ does not Granger-cause $j$.

---

[1]connections between two variables at the same time step

The shortcoming of this approach is that, when we have $S$ samples $\boldsymbol{x}_1, \ldots, \boldsymbol{x}_S$ with different underlying causal graphs, the parameters $\theta$ must be optimized separately for each of them. As a result, methods within this framework cannot take advantage of the information that might be shared between samples. This motivates us to question: can we amortize this process?

## 3 AMORTIZED CAUSAL DISCOVERY

We propose Amortized Causal Discovery (ACD), a framework in which we learn to infer causal relations across samples with different underlying causal graphs but shared dynamics. To illustrate: Suppose you want to infer synaptic connections (i.e. causal relations) between neurons based on their spiking behaviour. You are given a set of $S$ recordings (i.e. samples), each containing $N$ time-series representing the firing of $N$ individual neurons. Even though you might record across different populations of neurons with different wiring, the dynamics of how neurons connected by synapses influence one another stays the same. ACD takes advantage of such shared dynamics to improve the prediction of causal relations. It can be summarized as follows:

1. Define an encoding function $f_\phi$ which learns to infer Granger causal relations of any sample $\boldsymbol{x}_i$ in the training set $\boldsymbol{X}_{\text{train}}$. Define a decoding function $f_\theta$ which learns to predict the next time-step of the samples under the inferred causal relations.

2. Fit $f_\phi$ and $f_\theta$ to $\boldsymbol{X}_{\text{train}}$ by minimizing some loss $\mathcal{L}$: $f_{\phi_\star}, f_{\theta_\star} = \text{argmin}_{f_\phi, f_\theta} \mathcal{L}(\boldsymbol{X}_{\text{train}}, f_\phi, f_\theta)$.

3. For any given test sequence $\boldsymbol{x}_{\text{test}}$, simply output the Granger causal graph estimate $\hat{\mathcal{G}}_{\boldsymbol{x}_{\text{test}}}$:
   $\hat{\mathcal{G}}_{\boldsymbol{x}_{\text{test}}} = f_{\phi_\star}(\boldsymbol{x}_{\text{test}})$.

By dividing the model into two parts, an encoder and a decoder, ACD can use the *activations* of $f_{\phi_\star}$ to infer causal structure. This increases the flexibility of our approach greatly compared to methods that use the learned *weights* $\theta_\star$ such as the prior Granger causal discovery methods described in Section 2. In this section, we describe our framework in more detail, and provide a probabilistic implementation thereof. We also extend our approach to model hidden confounders.

**Preliminaries** We begin with a dataset $\boldsymbol{X} = \{\boldsymbol{x}_s\}_{s=1}^{S}$ of $S$ samples, where each sample $\boldsymbol{x}_s$ consists of $N$ stationary time-series $\boldsymbol{x}_s = \{\boldsymbol{x}_{s,1}, \ldots, \boldsymbol{x}_{s,N}\}$ across timesteps $t = \{1, ..., T\}$. We denote the $t$-th time-step of the $i$-th time-series of $\boldsymbol{x}_s$ as $\boldsymbol{x}_{s,i}^t$. We assume there is a directed acyclic graph $\mathcal{G}_s^{1:T} = \{\mathcal{V}_s^{1:T}, \mathcal{E}_s^{1:T}\}$ underlying the generative process of each sample. This is a structural causal model (SCM) (Pearl, 2009). Its endogenous (observed) variables are vertices $v_{s,i}^t \in \mathcal{V}_s^{1:T}$ for each time-series $i$ and each time-step $t$. Every set of incoming edges to an endogenous variable defines inputs to a deterministic function $g_{s,i}^t$ which determines that variable's value[2]. The edges are defined by ordered pairs of vertices $\mathcal{E}_s^{1:T} = \{(v_{s,i}^t, v_{s,j}^{t'})\}$, which we make two assumptions about:

1. No edges are instantaneous ($t = t'$) or go back in time. Thus, $t < t'$ for all edges.

2. Edges are invariant to time. Thus, if $(v_{s,i}^t, v_{s,j}^{t+k}) \in \mathcal{E}_s^{1:T}$, then $\forall 1 \leq t' \leq T - k : (v_{s,i}^{t'}, v_{s,j}^{t'+k}) \in \mathcal{E}_s^{1:T}$. The associated structural equations $g_{s,i}^t$ are invariant to time as well, i.e. $g_{s,i}^t = g_{s,i}^{t'} \forall t, t'$.

The first assumption states that causes temporally precede their effects and makes causal relations identifiable from observational data, when no hidden confounders are present (Peters et al., 2013; 2017, Theorem 10.1). The second simplifies modeling: it is a fairly general assumption which allows us to define dynamics that govern all time-steps (Eq. (2)).

Throughout this paper, we are interested in discovering the *summary graph* $\mathcal{G}_s = \{\mathcal{V}_s, \mathcal{E}_s\}$ (Peters et al., 2017). It consists of vertices $v_{s,i} \in \mathcal{V}_s$ for each time-series $i$ in sample $s$, and has directed edges whenever they exist in $\mathcal{E}_s^{1:T}$ at any time-step, i.e. $\mathcal{E}_s = \{(v_{s,i}, v_{s,j}) \mid \exists t, t' : (v_{s,i}^t, v_{s,j}^{t'}) \in \mathcal{E}_s^{1:T}\}$. Note that while $\mathcal{G}_s^{1:T}$ is acyclic (due to the first assumption above), the summary graph $\mathcal{G}_s$ may contain (self-)cycles.

**Amortized Causal Discovery** The key assumption for Amortized Causal Discovery is that there exists some fixed function $g$ that describes the dynamics of *all* samples $\boldsymbol{x}_s \in \boldsymbol{X}$ given their past

---

[2]The SCM also includes an exogenous (unobserved), independently-sampled error variable $\epsilon_v$ as a parent of each vertex $v$, which we do not model and thus leave out for brevity.

observations $\boldsymbol{x}_s^{\leq t}$ and their underlying causal graph $\mathcal{G}_s$:

$$\boldsymbol{x}_s^{t+1} = g(\boldsymbol{x}_s^{\leq t}, \mathcal{G}_s) + \boldsymbol{\varepsilon}_s^t \quad . \tag{2}$$

There are two variables in this data-generating process that we would like to model: the causal graph $\mathcal{G}_s$ that is specific to sample $\boldsymbol{x}_s$, and the dynamics $g$ that are shared across all samples. This separation between the causal graph and the dynamics allows us to divide our model accordingly: we introduce an amortized causal discovery encoder $f_\phi$ which learns to infer a causal graph $\mathcal{G}_s$ given the sample $\boldsymbol{x}_s$, and a dynamics decoder $f_\theta$ that learns to approximate $g$:

$$\boldsymbol{x}_s^{t+1} \approx f_\theta(\boldsymbol{x}_s^{\leq t}, f_\phi(\boldsymbol{x}_s)) \quad . \tag{3}$$

We formalize Amortized Causal Discovery (ACD) as follows. Let $\mathbb{G}$ be the domain of all possible summary graphs on $\boldsymbol{x}_s$: $\mathcal{G}_s \in \mathbb{G}$. Let $\mathbb{X}$ be the domain of any single step, partial or full, observed sequence: $\boldsymbol{x}_s^t, \boldsymbol{x}_s^{\leq t}, \boldsymbol{x}_s \in \mathbb{X}$. The model consists of two components: a causal discovery encoder $f_\phi : \mathbb{X} \to \mathbb{G}$ which infers a causal graph for each input sample, and a decoder $f_\theta : \mathbb{X} \times \mathbb{G} \to \mathbb{X}$ which models the dynamics. This model is optimized with a sample-wise loss $\ell : \mathbb{X} \times \mathbb{X} \to \mathbb{R}$ which scores how well the decoder models the true dynamics of $\boldsymbol{x}_s$, and a regularization term $r : \mathbb{G} \to \mathbb{R}$ on the inferred graphs. For example, this function $r$ may enforce sparsity by penalizing graphs with more edges. Note, that our formulation of the graph prediction problem is unsupervised: we do *not* have access to the true underlying graph $\mathcal{G}_s$. Then, given some dataset $\boldsymbol{X}_{\text{train}}$ with $S$ samples, we optimize:

$$f_{\phi_\star}, f_{\theta_\star} = \text{argmin}_{f_\phi, f_\theta} \mathcal{L}(\boldsymbol{X}_{\text{train}}, f_\phi, f_\theta) \tag{4}$$

$$\text{where } \mathcal{L}(\boldsymbol{X}_{\text{train}}, \phi, \theta) = \sum_{s=1}^{S} \sum_{t=1}^{T-1} \ell(\boldsymbol{x}_s^{t+1}, f_\theta(\boldsymbol{x}_s^{\leq t}, f_\phi(\boldsymbol{x}_s))) + r(f_\phi(\boldsymbol{x}_s)) \quad . \tag{5}$$

See Appendix B for a proof of the consistency of the loss $\ell$ and a discussion on regularization $r$.

Once we have completed optimization, we can perform causal graph prediction on any new input test sample $\boldsymbol{x}_{\text{test}}$ in two ways – we can feed $\boldsymbol{x}_{\text{test}}$ into the amortized encoder and take its output as the predicted edges (Eq. 6); or we can instantiate our estimate $\hat{\mathcal{G}}_{test} \in \mathbb{G}$ which will be our edge predictions, and find the edges which best explain the observed sequence $\boldsymbol{x}_{\text{test}}$ by minimizing the (learned) decoding loss with respect to $\hat{\mathcal{G}}_{test}$, which we term *Test-Time Adaptation (TTA)* (Eq. 7):

$$\hat{\mathcal{G}}^{\text{Enc}} = f_{\phi_\star}(\boldsymbol{x}_{\text{test}}) \quad ; \tag{6}$$

$$\hat{\mathcal{G}}^{\text{TTA}} = \text{argmin}_{\hat{\mathcal{G}}_{test} \in \mathbb{G}} \mathcal{L}(\boldsymbol{x}_{\text{test}}, \hat{\mathcal{G}}_{test}, f_{\theta_\star}) \quad . \tag{7}$$

By separating the prediction of causal relations from the modeling of their dynamics, ACD yields a number of benefits. ACD can learn to infer causal relations across samples with different underlying causal graphs, and it can infer causal relations in previously unseen test samples without refitting (Eq. (6)). By generalizing across samples, it can improve causal discovery performance with increasing training data size. We can replace either $f_\phi$ or $f_\theta$ with ground truth annotations, or simulate the outcome of counterfactual causal relations. Additionally, ACD can be applied in the standard causal discovery setting, where only a single causal graph underlies all samples, by replacing the amortized encoder $f_\phi$ with an estimated graph $\hat{\mathcal{G}}$ (or distribution over $\mathbb{G}$) in Eq. (4).

## 3.1 A Probabilistic Implementation of ACD

We take a probabilistic approach to ACD and model the functions $f_\phi$ and $f_\theta$ using variational inference (Fig. 2). We amortize the encoder $f_\phi$ with a function $q_\phi(\boldsymbol{z}|\boldsymbol{x})$, which outputs a distribution over $\boldsymbol{z}$ representing the predicted edges $\hat{\mathcal{E}}$ in the causal graph; and we learn a decoder $p_\theta(\boldsymbol{x}|\boldsymbol{z})$ which probabilistically models the dynamics of the time-series under the predicted causal relations. We choose a negative log-likelihood for the decoder loss $\ell$ and a KL-Divergence to a prior distribution over $\mathbb{G}$ for the regularizer $r$. As a result, our loss function $\mathcal{L}$ is a variational lower bound:

$$\mathcal{L} = \mathbb{E}_{q_\phi(\boldsymbol{z}|\boldsymbol{x})}[\log p_\theta(\boldsymbol{x}|\boldsymbol{z})] - \text{KL}[q_\phi(\boldsymbol{z}|\boldsymbol{x})||p(\boldsymbol{z})] \quad . \tag{8}$$

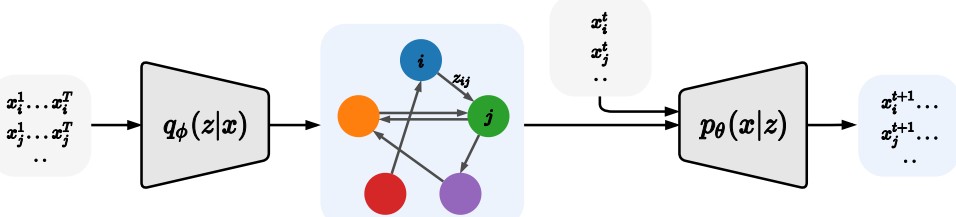

Figure 2: A Probabilistic Implementation of ACD. An amortized encoder $q_\phi(z|x)$ predicts the causal relations between the input time-series $x$. A decoder $p_\theta(x|z)$ learns to predict the next time-step of the time-series $x^{t+1}$ given their current values $x^t$ and the predicted relations $z$. This separation between causal relation prediction and modeling lets us train the model across samples with different underlying causal graphs but shared dynamics.

**Encoder** The encoder $q_\phi(z|x)$ applies a graph neural network $f_{\text{enc},\phi}$ (Gilmer et al., 2017; Kipf and Welling, 2017; Li et al., 2016; Scarselli et al., 2008) to the input, which propagates information across a fully connected graph $\mathcal{G} = \{\mathcal{V}, \mathcal{E}\}$. This graph includes vertices $v_i \in \mathcal{V}$ for each time-series $i$, and each pair of vertices $(v_i, v_j)$ is connected by an edge.

$$\boldsymbol{\psi}_{ij} = f_{\text{enc},\phi}(\boldsymbol{x})_{ij} \tag{9}$$

$$q_\phi(\boldsymbol{z}_{ij}|\boldsymbol{x}) = \texttt{Softmax}\left(\boldsymbol{\psi}_{ij} \,/\, \tau\right) \; . \tag{10}$$

To enable us to backpropagate through the samples of the discrete distribution $q_\phi(\boldsymbol{z}_{ij}|\boldsymbol{x})$, during training, we relax it by adding Gumbel distributed noise $\boldsymbol{g}$ (Jang et al., 2017; Maddison et al., 2017):

$$\boldsymbol{z}_{ij} \sim \texttt{Softmax}\left((\boldsymbol{\psi}_{ij} + \boldsymbol{g}) \,/\, \tau\right) \; . \tag{11}$$

The output $\boldsymbol{z}_{ij}$ of the encoder represents the predicted edges $\hat{\mathcal{E}}_{\text{Enc}}$ in the causal graph $\hat{\mathcal{G}}_{\text{Enc}}$. We consider the possibility that there are $n_\mathcal{E}$ different edge-types expressing causal relationships; for instance, inhibitory or excitatory synaptic connections. Then, more specifically, $z_{ij,e} = 1$ expresses that there is a directed edge of type $e$ from time-series $i$ to $j$, where $e \in \{0, \ldots, n_\mathcal{E} - 1\}$.

**Decoder** The decoder $p_\theta(x|z)$ models the dynamics of the time-series under the predicted causal relations. It uses both the predicted causal relations $\boldsymbol{z}_{ij}$ and the feature vectors of the time-series at the current time-step $t$, $\boldsymbol{x}^t = \{\boldsymbol{x}_1^t, \ldots \boldsymbol{x}_N^t\}$ as its input. First, it propagates information along the predicted edges by applying a neural network $f_e$, using the zero function for $f_0$:

$$\boldsymbol{h}_{ij}^t = \sum_{e>0} z_{ij,e} f_e([\boldsymbol{x}_i^t, \boldsymbol{x}_j^t]) \quad . \tag{12}$$

Then, the decoder accumulates the incoming messages to each node and applies a neural network $f_v$ to predict the change between the current and the next time-step:

$$\boldsymbol{\mu}_j^{t+1} = \boldsymbol{x}_j^t + f_v\left(\left[\sum_{i\neq j} \boldsymbol{h}_{ij}^t, \boldsymbol{x}_j^t\right]\right) \tag{13}$$

$$p_\theta(\boldsymbol{x}_j^{t+1}|\boldsymbol{x}^t, \boldsymbol{z}) = \mathcal{N}(\boldsymbol{\mu}_j^{t+1}, \sigma^2\mathbb{I}) \quad . \tag{14}$$

In other words, the decoder predicts $\Delta\hat{\boldsymbol{x}}^t$, which is added to the current value of the time-series to yield the prediction for the next time-step $\hat{\boldsymbol{x}}^{t+1} = \boldsymbol{x}^t + \Delta\hat{\boldsymbol{x}}^t$.

**Prediction of Causal Relations** In order to align our model with the philosophy of Granger Causality, we include a "no edge"-type edge function: If the encoder predicts the "no edge"-type edge $e = 0$ by setting $z_{ij,0} = 1$, the decoder uses the zero function and no information is propagated from time-series $i$ to $j$ (Eq. (12)). Due to this, time-series $i$ will Granger cause the decoder-predicted time-series $j$ only when the edge is predicted to exist (see Appendix A). Hence, by the same logic that justifies prior Granger causal work (Section 2), we expect the predicted edges to correspond to Granger causal relations. Finally, since we assume no hidden confounders and no instantaneous edges, these Granger causal relations will correspond to relations in the underlying SCM (Peters et al., 2017, Theorem 10.3).

### 3.2 HIDDEN CONFOUNDING

Hidden confounders are a critical problem in the time-series context: when they exist, Granger causality is not guaranteed to correspond to the true causal graph anymore (Peters et al., 2017, Theorem 10.3)[3]. Inspired by proxy-based methods from causal inference (e.g. Louizos et al. (2017), see Section 4), we present a method for applying ACD to the hidden confounding setting. First, we extend the amortized encoder $q_\phi(z|x)$ to predict an additional variable. Then, we encourage this variable to model the hidden confounder by applying a structural bias – depending on the type of unobserved variable that we want to model, its predicted value is utilized differently by the remaining model. The decoder remains responsible for modeling the dynamics, and now also processes the predictions for the unobserved variable. While this setup might not allow us to identify the hidden confounders, the data-driven approach underlying ACD can benefit our model: by pooling the statistical strength across samples with different underlying causal graphs, our model can learn to mitigate the effects of the hidden confounders.

We consider two types of hidden confounders: 1) a temperature variable that confounds all endogenous variables by influencing the strength of their causal relations. This temperature is sampled separately for each sample, and remains constant throughout each sample. 2) a hidden variable that behaves just like the observed variables, i.e. it may affect or be affected by the observed variables through the same causal relations, and its value changes across time. In both cases, we extend the encoder to predict this hidden variable, and feed that prediction into the decoder. We provide more details in Section 5.2.

## 4 RELATED WORK

A range of approaches to causal discovery in both temporal and non-temporal data exist (Heinze-Deml et al., 2018; Peters et al., 2017; Spirtes et al., 2000). One common class is *constraint-based*, relying on conditional independence testing to uncover an underlying DAG structure or equivalence class (Spirtes et al., 2000). These methods predict a single graph $\hat{\mathcal{G}}$ (or equivalence class) for all samples. There is no notion of fitting a dynamics model for time-series methods in this class (Entner and Hoyer, 2010). Another common class of methods for causal discovery is *score-based* (Bengio et al., 2019; Chickering, 2002). Here, a score function $h$ is chosen, and the methods perform a search through graph space to optimize this score, i.e. $\hat{\mathcal{G}} = \mathrm{argmin}_{\mathcal{G}} \, h(\mathcal{G})$. Our proposed decoder-based inference (Eq. (7)) can be seen as score-based causal discovery with a *learned* score function $\mathcal{L} \circ f_{\theta_\star}$. A third class of methods fits a (possibly regularized) dynamics model $f$ and then analyzes its form to produce a causal graph estimate, by using linear dynamics (Hyvärinen et al., 2010), recurrent models (Khanna and Tan, 2020; Nauta et al., 2019; Tank et al., 2018), or other deep-learning based approaches (Lachapelle et al., 2019; Wu et al., 2020; Zheng et al., 2020). See Section 2 for discussion. Other approaches to causal discovery in temporal data use independence or additivity assumptions (Eichler, 2012; Peters et al., 2013). A number of works have explored the idea of jointly learned causal structure across examples, including a range of papers in the setting where a number of related datasets are collected, possibly with different columns (Dhir and Lee, 2020; Huang et al., 2019; 2020; Shimizu, 2012; Tillman and Eberhardt, 2014). Li et al. (2018) proposes learning a linear mixed effects model across samples, and concurrent work explores amortized deep learning of differing types of causal structure (Ke et al., 2020; Li et al., 2020). Very few papers systematically study the hidden confounding setting. Some empirical work shows that encoder-based models with enough proxies (variables caused by hidden confounders) can improve causal inference under hidden confounding (Louizos et al., 2017; Parbhoo et al., 2020), and theoretical work proves the identifiability of latent variables from proxies under some assumptions (Allman et al., 2009; Kruskal, 1977).

Several works have used graph neural networks (Battaglia et al., 2016; Kipf et al., 2018; Santoro et al., 2017) or attention mechanisms (Fuchs et al., 2019; Goyal et al., 2019; Van Steenkiste et al., 2018; Vaswani et al., 2017) to infer relations between time-series. Alet et al. (2019) propose a meta-learning algorithm to additionally model unobserved variables. While these approaches model object relations in a number of ways, they are not explicitly designed to infer *causal* graphical structure.

---

[3]For instance, if an unobserved time-series $U$ causes both time-series $X$ and $Y$, then the past of $X$ can help predict the future of $Y$, even though there is no causal link between them.

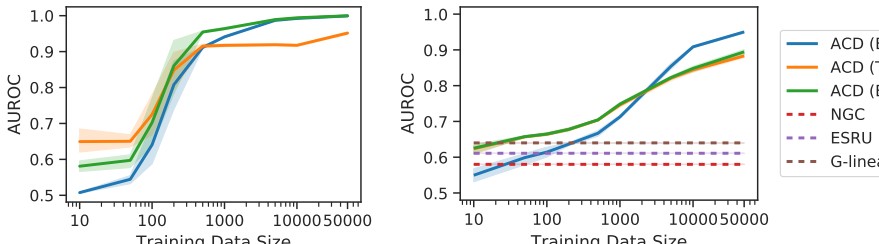

Figure 3: Causal discovery performance (in AUROC) on the particles dataset (A-left) and Kuramoto (B-right). ACD improves with more training data, outperforming previous approaches with as few as 50 available training samples on Kuramoto. In the high-data regime, encoder inference (*Enc*) is best, while test-time adaptation (*TTA* and *Enc+TTA*) is superior in low-data settings.

The probabilistic implementation of ACD is based on the Neural Relational Inference (NRI) model (Kipf et al., 2018), extended by a special zero-function to allow for a causal interpretation of the inferred edges, new inference methods using test-time adaptation and a new algorithm to handle confounders. Moreover, the new model is applied to a different problem than NRI, namely (Granger) causal discovery, and we show that it outperforms the current state of the art for this type of problem.

## 5 EXPERIMENTS

**Implementation**  We measure causal discovery performance by area under the receiver operator curve (AUROC) of predicted edge probabilities over test samples. We compare to recurrent models (Khanna and Tan (2020); Tank et al. (2018)), and a mutual-information (MI) based model by Wu et al. (2020) and several baselines implemented by those authors, including MI (unmodified), transfer entropy (Schreiber, 2000), and a linear Granger causality. More details in Appendix C.

### 5.1 FULLY OBSERVED AMORTIZED CAUSAL DISCOVERY

We test ACD on three datasets: two fully-observed physics simulations (Kuramoto and Particles) and the Netsim dataset of simulated fMRI data (Smith et al., 2011). Note, in contrast to the physics simulations used in Kipf et al. (2018), we generate data with *asymmetric* connectivity matrices to represent causal relations.

First, we test our method on the **Kuramoto dataset**, which contains five 1-D time-series of phase-coupled oscillators (Kuramoto, 1975). We find that ACD greatly outperforms all approaches for Granger causal discovery that we compare against (Table 1). In contrast to these approaches, ACD achieves this result *without* fitting to the test samples. Additionally, we find that ACD can indeed utilize samples with different underlying causal graphs – its performance improves steadily with increasing training data size (Fig. 3). Nonetheless, it is also applicable to the low-data regime: when applying ACD

| Method | AUROC |
|---|---|
| MPIR (Wu et al., 2020) | $0.502 \pm 0.006$ |
| Transfer Entropy (Schreiber, 2000) | $0.560 \pm 0.005$ |
| NGC (Tank et al., 2018) | $0.574 \pm 0.018$ |
| eSRU (Khanna and Tan, 2020) | $0.607 \pm 0.001$ |
| Mutual Information | $0.616 \pm 0.000$ |
| Linear Granger Causality | $0.647 \pm 0.003$ |
| Amortized Causal Discovery | $\mathbf{0.952 \pm 0.003}$ |

Table 1: AUROC for causal discovery on Kuramoto dataset. 95% confidence interval shown.

with test-time adaptation (TTA), it requires less than 50 training samples to outperform all previous approaches. We note that the baseline performance here is worse than presented elsewhere in the literature — this is because we evaluate accuracy without considering prediction of self-connectivity, as self-connections are the easiest type to predict.

In our second experiment, we apply ACD to the **particles dataset**. This dataset models five particles that move around a two-dimensional space, with some particles influencing others uni-directionally by pulling them with a spring. Since all previous methods were intended for one-dimensional time-series,

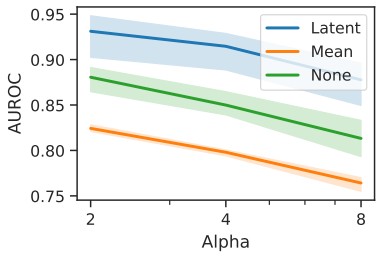

Figure 4: AUROC with unobserved temperature. ACD with a *latent* variable outperforms a baseline which imputes a *mean* temperature, and a learned fixed-temperature decoder (*None*).

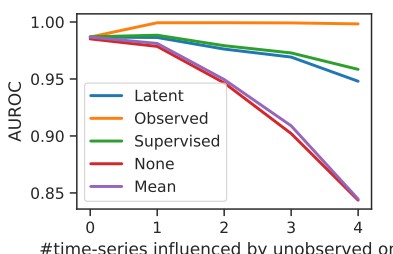

Figure 5: AUROC with unobserved time-series. As more time-series are influenced by the unobserved one (x-axis), the benefit of using an additional *latent* variable for modeling its effects grows.

we were unable to evaluate them in this domain. ACD, on the other hand, is readily applicable to higher-dimensional data, and performs almost perfectly on this dataset with **0.999 AUROC**.

In both experiments, causal relation prediction with the learned encoder (Enc - Eq. (6)) performs best in the high-data regime, while test-time adaptation (TTA - Eq. (7)) improves the performance in low-data settings (Fig. 3). This benefit of TTA can be largely attributed to two effects. First, TTA closes the amortization gap of the encoder (Cremer et al., 2018). Second, TTA overcomes the encoder's overfitting on the training data (as seen in the training curves in Appendix C.2) by being adapted to the individual test samples. On the particles dataset, initializing TTA with the encoder's prediction (Enc+TTA) improves over a random initialization (TTA) as the encoder improves; but we do not observe this effect on the Kuramoto dataset.

Finally, we apply ACD to the **Netsim** dataset (Smith et al., 2011) of simulated fMRI data. Here, the task is to infer the underlying connectivity between 15 brain regions across 50 samples. A single graph underlies all samples, allowing us to demonstrate ACD's applicability to the classical setting. We replace the amortized encoder $q_\phi(z|x)$ with a global latent distribution $q(z)$, optimize it through the decoder, and then use test-time adaptation (TTA). Even though our model cannot benefit from its data-driven design here, it performs comparably to methods that are intended for use in the single-graph setting (Table 2).

| Method | AUROC |
|---|---|
| MPIR (Wu et al., 2020) | $0.484 \pm 0.017$ |
| Transfer Entropy (Schreiber, 2000) | $0.543 \pm 0.003$ |
| NGC (Tank et al., 2018) | $0.624 \pm 0.020$ |
| eSRU (Khanna and Tan, 2020) | $0.670 \pm 0.015$ |
| Mutual Information | $\mathbf{0.728 \pm 0.002}$ |
| Linear Granger Causality | $0.503 \pm 0.004$ |
| Amortized Causal Discovery | $0.688 \pm 0.051$ |

Table 2: AUROC for causal discovery on Netsim dataset. 95% confidence interval shown.

## 5.2 Amortized Causal Discovery under Hidden Confounding

### 5.2.1 Latent Temperature

In this experiment, we use the particles dataset and vary an unobserved temperature variable, which modulates how strongly the particles exert force on each other – higher temperatures result in stronger forces and a more chaotic system. For each $x_s$, we sample an independent temperature $c \sim \text{Categorical}([\frac{\alpha}{2}, \alpha, 2\alpha])$ from a categorical distribution with $\alpha \in \mathbb{R}$ and equal probabilities. We predict this unobserved temperature by extending the amortized encoder with an additional latent variable which models a uniform distribution. Then, we add a KL-Divergence between this posterior and a uniform prior on the interval $[0, 4\alpha]$ to our variational loss. To allow for learning in this setting, we introduce an inductive bias: we use a decoder which matches the true dynamics $g$ given the predicted temperature and causal relations. See Appendix D.1 for more details and additional results.

**Results** Fig. 4 shows the causal discovery results across different values of $\alpha$. ACD enhanced with an additional latent variable (*Latent*) outperforms both tested baselines across all temperatures: *Mean*, which uses the same ground-truth decoder as *Latent* and fixes the decoder temperature to be

the mean of the categorical distribution, and *None*, which does not model $c$ explicitly and trains an MLP decoder. Additionally, this method achieves high predictive performance on the unobserved temperature variable: for $\alpha = 2$, temperature prediction obtains 0.888 $R^2$, 0.966 AUROC and 0.644 accuracy. These results indicate that we can model an unobserved temperature variable, and thus improve robustness under hidden confounding.

### 5.2.2 UNOBSERVED TIME-SERIES

Here, we treat one of the original time-series in the particles dataset as unobserved. It exhibits the same dynamics as the observed time-series, evolving and causally influencing others the same way as before. This challenging setting has received little attention in the literature; Alet et al. (2019) tackled it with mixed success. We model the unobserved time-series by extending the amortized encoder with an additional latent variable and applying a suitable structural bias: the latent prediction $z_u^t$ for time-steps $t = \{1, ..., T\}$ is treated in the same way as the observed time-series $x$. Its entire trajectory is used by the encoder to predict causal relations, and its value at the current time-step is fed into the decoder. See Appendix D.2 for more details and additional results.

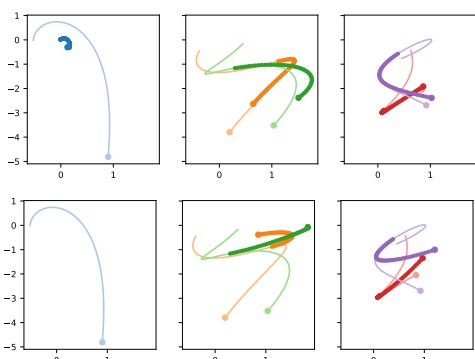

Figure 6: Trajectory prediction with an unobserved time-series (TS). Faded: ground truth. Bold: prediction, starts after observing the first half of the ground truth. Dots denote end of TS. Top: ACD with *Latent*, bottom: *None* baseline - does not model unobserved TS. Left: unobserved TS, middle: TS directly influenced by unobserved, right: remaining TS. Though we underestimate the unobserved TS, observed TS prediction improves.

**Results** Fig. 5 shows how the causal discovery AUROC depends on the number of observed time-series directly influenced by the unobserved one. When this number is zero, all tested approaches perform the same. With growing numbers of influenced time-series, the baselines that either ignore the missing time-series (*None*) or impute its value with the average of the observed time-series over time (*Mean*) deteriorate strongly. In contrast, the proposed ACD with a *Latent* variable stays closer to the performance of the fully *Observed* baseline. As shown in Fig. 6, it also improves the future trajectory prediction of the observed time-series. A *Supervised* baseline that uses the (usually unavailable) ground-truth trajectories to optimize the prediction of the unobserved time-series, improves only slightly over our approach. These results indicate that ACD can use latent variables to improve robustness to unobserved time-series.

## 6 CONCLUSION

In this paper, we introduce ACD, a framework for causal discovery in time-series data which can leverage the information that is shared across samples. We provide a probabilistic implementation and demonstrate significant performance gains when predicting causal relations, even under hidden confounding. Exciting future directions include interventions, more flexible graph structures, or methods that adapt dynamically to the type of hidden confounder at hand.

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
