# OpenReview forum: "Amortized Causal Discovery: Learning to Infer Causal Graphs from Time-Series Data"
_ICLR.cc/2021/Conference — Reject_

### Official Review · AnonReviewer3 · 2020-10-19
**The authors leverage the potential information shared across samples to illicit causal discovery.**

**Rating:** 5
**Confidence:** 4

**Review:**

# Introduction

You should have praise for your first sentence in your second paragraph. It is excellent. A neat summary indeed. Emphasise it if you can.

Refer to figure one in paragraph one to allow the novice reader to get a quick understanding of what you are trying to do. Better yet, move figure one into the first page and put it at the bottom, it is a great eyecatcher.

# Background

N should be italicised in definition 2.1.

# Amortized causal discovery

Presumably like all work of this kind you assume, there are no cycles or self-cycles in the SCM. That is pretty common fare for causal work (and a huge limitation) but it should still be noted at the very top of this section or indeed expanded upon in the preliminaries. I take it thus that you cannot handle SCMs with self-cycles and cycles in the endogenous variables? A far harder problem of course, but just to be clear from this reviewer's point of view.

What your second point is saying in the preliminaries is that you assume that your data is sampled from a stationary process. You may want to add that key-word as that says a lot to people from outside this field, and again, emphasises the limitations of this causal discovery (and most in fact) method. It also gives you plenty of scope for future work! Indeed you go on to talk about the summary graph which is by definition only defined for stationary processes.

A common theme throughout this paper is that you just insert figures without referring to them beforehand. This is very confusing. Please stop doing this and re-write the paper so that your figures follow the discussion of them. You introduce an unnecessary break in the flow of the reading otherwise, like between pages 1 and 2 and 3 and 4.

Please explain what you mean by this: "Note, that we do not apply a supervised loss on the graph prediction"

Use \text{} in your subscripts in your equations, it reads better i.e. $\hat{\mathcal{G}}_{\text{TTA}}$ is preferable.

Please explain what you mean by "expresses that there is a directed edge of type e from time-series i to j" -- what is type e?

## Hidden confounding

This section could do with a diagram. I find it very confusing. I think you are saying that you allow the decoder to predict the additional variable (in the summary graph?) which confounds _all_ endogenous variables in the summary? Or some? or one? Which is it? What if the confounding variable does not affect all endogenous variables?

# Related work

Solid section. Well done.

# Experiments

## 5.1

It would be helpful if you could show the inferred adjacency matrix of the particle dataset, alongside the GT adjacency matrix.

These experiments feel very selective. Your goal is to demonstrate ACD's superiority over the other methods, granted, that's the process and we all know it. However, this does seem like an odd comparison. Take Khanna and Tan, 2020 (not 2019 like you've quoted it): on a Lorenz-96 simulation, they reach an average AUROC of 0.95. Yet here they get 0.607, and this ought to be because you have generated asymmetric connectivity matrices which renders a structure which eSRU cannot handle. But how we do know? You haven't run ACD on say the Lorenz-96 system which is pretty common fare in causal discovery. If your model does not outperform SOTA methods in that space you may have to relax some of your claims. What is more, Tank et al do very well on Lorenz-96 too. They are not particularly good at the DREAM-3IN SILICO NETWORK INFERENCE CHALLENGE, and that's where ACD could shine through, rather than taking these models 'out of context' as it were. Point being, please demonstrate ACD on the _simpler_ operating space where the likes of NGC and eSRU live. You appear to have started that process with Table 2 but that is a complex setting, perhaps too complex to understand where ACD is failing. Again, I advise starting simple with the standard Lorenz-96 model just to ensure that your method performs well in that easy space.

# 5.1

It is not okay for you to bold ACD's performance value in table 2. Remove it. MI is clearly higher and putting anything else is bold is confusing.

## 5.2.2

This experiment is very interesting. The figures are good too. My advice would be to make the axis text larger, and set alpha=0.75 or something when you plot the prediction -- that way the reader can better see how the prediction tracks the GT.

However, I am a bit confused about the deterministic chaos that you briefly mention at the start of 5.2.1 -- the parameter setting you use in this section (5.2.2) presumably does not give rise to deterministic chaos?

---

None of your experiments contains fully identifiable noise models from what I can tell? You mention "noise" twice in the manuscript. None is in the experimental section. Now since you are only using simulations, I find it odd that you do not discuss this nor ACD's robustness against noise. As you are fully aware of course, most of these models break-down in the presence of a low SNR. And for this to be useful at all in the 'real' world, how does ACD fare?

---

> ### Author Response · Authors · 2020-11-16
> **Responses on confounding, experiments, and more (Part 1)**
>
> We thank the reviewer for their helpful feedback. We have updated our paper to incorporate a number of the proposed changes (e.g. referencing all figures, using \text{} subscripts, as well as the changes discussed below).
>
> Additionally, we would like to answer questions and address concerns raised in the review:
>
>
> ## Cycles in the SCM
> In the summary graph, our approach can handle cycles and assumes self-cycles for all variables. The graph that is unrolled over time, on the other hand, we assume to be a directed acyclic graph due to our assumptions of non-instantaneous effects, and no effects going backwards in time. We have added a clarification on this to the paper.
>
> ## Stationarity
> We do mention the key-word stationary in two places already: 1) in definition 2.1 (Non-linear Granger Causality) and 2) in the first sentence of the preliminaries.
>
> ## Supervised Loss
> With the sentence "Note, that we do not apply a supervised loss on the graph prediction", we simply wanted to highlight that our approach learns to infer causal relations in an unsupervised way. We clarified this in the updated paper.
>
> ## Edge type e
> In our approach, we have one fixed edge-type, edge 0, that represents that there is no causal relation between two variables. If there is a causal relation between two variables, this relation might manifest through different dynamics that our model can learn to handle through different edge-types $e$. For example, if we were to model how neurons react to the activities of other neurons, we could have two separate edge types to represent excitatory and inhibitory connections between neurons. We have added a clarification on this to the paper.
>
> ## Hidden confounding
> Indeed, we model hidden confounders differently depending on their type. In section 5.2.1, we vary an unobserved temperature variable, which modulates how strongly the particles exert force on each other. In this case, the hidden confounder may confound all endogenous variables - specifically, through strengthening (or weakening) the force of every causal connection, it will alter the movement of particles which are affected by other particles. Thus, we integrate the model’s prediction for that variable into the model in such a way that it may influence the future prediction of all variables in the decoder.
>
> In section 5.2.2, we treat one of the original time-series in the particles dataset as unobserved. In this case, the hidden confounder may confound all, some or none of the endogenous variables (as depicted on the x-axis of figure 5) and the model has to learn to differentiate between these cases. The reverse might also hold, i.e. the observed variables may influence the unobserved one, making this a very challenging setting.
>
> We have clarified this distinction in the updated paper.
>
> ## Depicting the adjacency matrix
> We show exemplary GT causal graphs in Figure 1 (the graphs on the right correspond to the samples shown on the left). Since we achieve an AUROC of 0.999, the inferred causal graphs will correspond to these in the overwhelming number of cases.

---

> ### Author Response · Authors · 2020-11-16
> **Responses on confounding, experiments, and more (Part 2)**
>
> (continued from part 1)
> ## Experiment selection
> We agree that the comparison between ACD and previous methods can appear odd. However, we believe that this is due to the nature of ACD, its applicability to and superiority in previously unexplored settings. ACD shines on datasets with samples with different underlying causal graphs (e.g. Particles and Kuramoto), as it can leverage the information that is shared between samples. We believe that this is the main reason for it to outperform previous methods so strongly on Kuramoto. If anything, the comparison on this dataset (table 1) is actually skewed towards previous methods as they are trained and evaluated on the test samples, while ACD simply predicts the causal relations in the test samples without training on them.
>
> We see two reasons as to why previous methods perform worse in our experiments on Kuramoto compared to Lorenz-96: (1) The samples in Kuramoto are relatively short (49 time-steps, compared to 250/500/1000 used in Lorenz-96 in the literature), making it harder to infer the correct causal graph solely based on a single sequence. (2) In our model, we assume self-connectivity for all time-series and only evaluate the causal discovery performance between different time-series. This is in contrast to the evaluation shown in the literature, which also evaluates the arguably easier task of whether self-connectivity is predicted correctly. On Netsim, for example, this factor alone leads to a performance drop of more than 15 percent points when comparing the evaluation of previous methods presented in Khanna et al (2020) to ours.
>
> Finally, we do not believe that running ACD on Lorenz-96 would yield new insights. In particular, we believe that the fact that Lorenz-96 only contains a single causal graph invites overfitting through hyperparameter tuning, which we observe in some previous work. eSRU, for example, has optimized some of their hyperparameters up until 6 digits after the comma. Two of their bias terms, for example, are set to be $mu_1 = 0.021544$ and $mu_2 = 0.031623$ (see https://github.com/sakhanna/SRU_for_GCI). As a result, we believe that a comparison on Lorenz-96 is more likely to show a lack in hyperparameter tuning than an actual superiority of one method over another.
>
> Note, that Netsim has received a similar treatment of their hyperparameters in the literature (same link above). Nonetheless, our method achieves comparable results to these methods with considerably less hyperparameter tuning. This is despite the fact that this dataset is actually taking ACD “out of context” by solely containing samples with a single underlying causal graph.

---

### Official Review · AnonReviewer1 · 2020-10-25
**Interesting idea, but lacking investigation into identifiablity and consistency**

**Rating:** 6
**Confidence:** 4

**Review:**

This paper studies the problem of learning causal graphs from time-series data. A new framework, called Amortized Causal Discovery (ACD), is proposed based on the observation that samples often share identical or similar dynamics. ACD is then implemented with a variational model, mostly following existing works. Overall, I think it is a very interesting idea to consider shared dynamics and, in some sense, to 'learn a score function' from data, compared with existing score-based methods that usually need to put assumptions on data generations to make the score function sound. However, there is also a key problem, identifibility and consistency, to ACD, for which I do not recommend an acceptance for now. I would be happy to increase my score if authors can address my concerns.

Identifiability and Consistency: (a) identifiability is an important problem to causal discovery from observational data. I find that only Peters et al, 2013 is cited for such an issue. However, in that work, the identifibility result requires a much stronger condition than the assumption used in this paper. It would be good to discuss this issue in more details. (b) as stated in Section, 4, the proposed approach can be seen a score based method with learned score functions. However, it is unclear when this score function could be 'consistent', i.e., maximizing the score function would lead to a consistent estimate of the true graph (possibly under some model assumptions). (c) Another part, related to this learned score function, is whether the overall approach is consistent. In particular, Eq. (5) has a regularization term (which is often a sparsity term, as stated in the paper). The question is how to pick this penalty term, including its weight, in practice? This is crucial for causal discovery and I do not find that this issue is explained in the current version.

Writing: the overall writing is very good, but could be improved for some places. (a) Eq (12) uses $z_{ij, 0}$ which was not introduced until the next paragraph; (b) it seems that the problem size (how many variables of the problem) is small. It would be good to introduce the problem size explicitly in the main content. (c) as mentioned above, the identifiably and regularization term are not clearly discussed; they are however important to score based causal discovery.

Other questions: (a) why the BIC score used by Chickering 2012 is called 'heuristic score'? (b) Gumbel technique may not give exactly zero or one, so how was the 'no edge' type processed? This part was somewhat omitted. (c) what if there are instantaneous effects? I do not see why ACD does not apply to this case.

*********after reading rebuttal *******
I have increased my rating, but may still have some concerns. See below.

---

> ### Author Response · Authors · 2020-11-18
> **Responses on consistency, identifiability, and more**
>
> We thank the reviewer for their thoughtful feedback. We have updated our paper to incorporate the proposed changes. Most importantly, we have included a proof of consistency and a discussion on the regularization term in the appendix.
>
> Additionally, we would like to answer questions and address concerns raised in the review:
>
> Identifiability and Consistency:
> (a) We would like to clarify our identifiability citation. The reviewer is correct that Peters et al 2013 requires a stronger assumption than we make - in particular, Theorem 1, condition 2 states that the summary graph must be acyclic. However, in the proof of this theorem, it becomes clear that this assumption is only necessary if there are instantaneous effects. The correct citation for this is probably Peters et al, 2017, Theorem 10.1, which cites the 2013 paper, but makes the identifiability statement under the “no instantaneous effects” assumption directly. We have updated this citation. Then, the Peters et al 2017 citation which connects Granger causality to structural identification is Theorem 10.3 - we also clarified this in the updated paper.
>
> (b/c) We have included a proof for the consistency of our approach and a discussion on the regularization term in the updated version of our paper. We plan on improving the clarity of the proof when given more time for the camera-ready version by, e.g., adding examples to the arguments we make.
>
> We agree that a number of formulations for the regularization term in ACD are possible, and we only explore one in this paper due to the relationship with variational learning. We do believe that a sparsity-inducing term is valuable. Without any incentive to produce sparse graphs, the model will have little incentive to infer anything other than a complete graph. As with any regularization term, choosing the hyperparameter is a process of cross-validation. Within an order of magnitude, we found results weren’t too sensitive. It will depend ultimately on your confidence in your prior - if you are very uncertain about the number of edges in the graph the weight will be weaker.
>
> Writing:
> (a) We note that $z_{ij,e}$ is defined above the use of $z_{ij,0}$ in Eq 12  as an edge of type e between time series i and j. We have clarified this notation in the updated version.
> (b) We discuss the problem size in Appendix B.1.2 (5 for particles/Kuramoto and 15 for Netsim). However, we have added it to the main content in the updated version.
>
> Other questions:
> (a) We have changed the wording around the Chickering BIC score as calling it a “heuristic” may be underselling the value of its theoretical motivations
> (b) During training, $z_{ij}$ is indeed not a one-hot vector due to the Gumbel softmax, but a probability distribution over the different edge-types. In this case, we weigh the different edge functions by their respective probability, including the no-edge function (i.e. zero). We have updated equation 12 to reflect this.
> (c) The assumption of no instantaneous effects is required, according to Peters et al, 2017, Theorem 10.1 & 3, to make the full-time graph identifiable through Granger causality.

---

> > ### Comment · AnonReviewer1 · 2020-11-23
> > **Thanks for a detailed response. Some questions remain.**
> >
> > I appreciate the response from authors, particularly for attempting to address the identifiability and consistency. I have increased my rating but may still have some concerns, mainly for the consistency part in appendix.
> >
> > - right after Eq. (15): 'some loss function which is minimized at zero when its two inputs are equal'. Since the causal relationship has certain unknown, additive noise, minimizing a loss to zero implies overfitting and in practice, how can we make this assumption true for all possible values?
> > - the second assumption on monotonicity of $g$ seems strong.
> > - rather, a typical way to avoiding the possibility of spurious edges is to add a sparsity penalty, where the penalty weight is critical to both theoretical and empirical results. Besides, the regularization part may be further improved by putting the implementation details of the proposed method.
> >
> > Lastly, if one puts additional assumptions like Peters et al, 2013 so that identifiability is guaranteed, can the proposed method also include instantaneous cases?

---

> > > ### Author Response · Authors · 2020-11-24
> > > **Responses on noise, monotonicity and more**
> > >
> > > Thank you for the response - we are glad that we were able to address some of your concerns. In response to your most recent questions:
> > >
> > > * In our proof, there is no real notion of overfitting since we assume that the correct dynamics are already known, and we only consider a single test sample. However, we can consider a notion of robustness to noise by asking what would happen when this test sequence approaches infinite length:
> > >
> > >     Suppose that we have noise $\epsilon_t$ which is independently added to each time step $t$, with mean zero. For a sequence of length T, we have a MSE loss equal to $\sum_{t=1}^T [ g(x_t, \mathcal{G}) + \epsilon_t  - g(x_t, \mathcal{\hat{G}}) ]^2 = \sum_{t=1}^T [ g(x_t, \mathcal{G})  - g(x_t, \mathcal{\hat{G}}) ]^2  + \epsilon_t^2 + 2 \epsilon_t [ g(x_t, \mathcal{G})  - g(x_t, \mathcal{\hat{G}}) ]$.
> > >
> > >     As $T \rightarrow \infty$, this sum approaches its expectation, and with this the final term will go to 0 since $E[\epsilon_t] = 0$. Additionally, the noise is independent of the observations, i.e. ${E}\big[( \epsilon^t)^2\big]$ is constant. Therefore, even in the noisy case, a graph estimate $\hat{G}$ minimizes the MSE in the limit only when it minimizes $\sum_{t=1}^T [ g(x_t, \mathcal{G})  - g(x_t, \mathcal{\hat{G}}) ]^2$. We know that $\mathcal{\hat{G}} = \mathcal{G}$ minimizes this term at 0, and by the proof in the paper, the set of $x_t$ for which some other $\mathcal{\hat{G}} \neq \mathcal{G}$ also makes this 0 is a set of measure 0.
> > >
> > >     We include an argument along these lines in the updated version of the paper.
> > >
> > >
> > > * We agree that our monotonicity assumption may not hold in every scenario. We have changed its definition slightly in the updated version of the paper to be less restrictive. Now, only a perturbation at a single step of a particular lag needs to yield monotonic behavior. With this formulation, achieving monotonicity is possible in a range of scenarios - for instance in the particles and Netsim datasets used in our experiments.
> > >
> > > * We agree that regularization helps with spurious edges by yielding a lower loss for predicting non-causal relations. However, this can only explain why the model might not predict too many edges, not too few. We will add more discussion surrounding our implementation details to the camera-ready version of the paper.
> > >
> > > * We wouldn’t expect our model to work for instantaneous cases even if the additional assumptions are guaranteed, due to the structure of our model. We train the model by optimizing its future prediction performance - if we were to model instantaneous effects this would involve having the same time-step as an input that we are trying to predict.

---

### Official Review · AnonReviewer4 · 2020-10-26
**Interesting experimental work but lacking some implementation details and theoretical supports**

**Rating:** 6
**Confidence:** 4

**Review:**

The authors proposed a framework called Amortized Causal Discovery (ACD) for recovering causal relationships in time series where samples are generated from models with different underlying causal graphs but shared dynamics. This framework is applicable in settings such as modeling neural spiking trains where the dynamics of how neurons react to the activities of other neurons remain the same. In the proposed framework, they considered a causal discovery encoder $f_{\theta}$, which tries to extract causal relationships and map it to a latent space. Moreover, there is a dynamic decoder $f_{\phi}$, which provides one-step predictions. The proposed architecture is similar to a variational auto-encoder and the encoder part is based on graph neural networks.

Although experimental results show that a considerable improvement in some dataset (even in the presence of a latent variable), there are some concerns about the contributions of this work and also its presentation:

1- About the two main assumptions: The authors considered two assumptions on the causal model of time series (no instantaneous effect and invariant causal graph in time). We know that both assumptions are not satisfied in many practical applications. For instance, we have instantaneous effects if the sampling rate is low. Moreover, most causal mechanisms are changing in time.  As an example, in (Pfister et al., 2019), this problem has been addressed by detecting change points. I am not sure whether that approach is promising or not but it is expected from an experimental work to handle a more general setting.

2- About the shared dynamic: Is it possible to check whether the assumption of shared dynamic is valid in the observed data? What if there are some small changes in the dynamics across different training samples.

3- In Eq. (12-14), it seems that we are using the values of other time series at only time $t$ to give a one-step prediction of a particular time series. Why do not we consider the history of previous values from time $1$ to $t$?

4- About handling latent confounders: As noted by the authors, recovering causal relationships is challenging in the presence of latent variables. Although experimental results show that the proposed method has better performance than baselines, it is still unclear whether there are some theoretical guarantees in recovering the correct causal graph using the proposed method.

5- The presentation of the paper could be improved:
*Please provide the exact definition of $\hat{\mathcal{G}}_x$, $\mathbb{G}$, $\mathbb{X}$, $r$,....
*Please explain the two methods (the amortized encoder and TTA) in more detail on page 4.

(Pfister et al., 2019) Pfister, Niklas, Peter Bühlmann, and Jonas Peters. "Invariant causal prediction for sequential data." Journal of the American Statistical Association 114.527 (2019): 1264-1276.

---

> ### Author Response · Authors · 2020-11-16
> **Responses on instantaneous effects, shared dynamics, and more**
>
> We thank the reviewer for their helpful feedback and would like to address the concerns raised in the review:
>
> 1 - The assumption of no instantaneous effects allows us to interpret the Granger causal relations inferred by our approach as causal relations in the underlying SCM (Peters et al., 2017). More importantly, it is fundamental to the concept of Granger causality itself.
> We believe that we have made a significant contribution by working on a non-linear extension of Granger causality, and leave an extension for instantaneous effects for future work.
>
> The approach presented by Pfister et al., 2019 might indeed be applicable in our setting to lift the assumption of an invariant causal graph in time. However, as this is not the focus of our paper, we consider this to be an interesting avenue for future work.
>
> 2 - Within the scope of our paper, we assume to have prior knowledge about the observed data having shared dynamics.
> Nonetheless, it is possible to test this assumption informally. Only if the samples actually share their underlying dynamics, would we expect our approach of training a single model across samples to achieve a better performance. Thus, we could simply compare the same architecture trained in two ways: (1) by training an amortized encoder and shared decoder across all samples and (2) by optimizing a separate latent distribution q(z) and decoder for each individual sample. If (1) indeed achieves a better performance, we would conclude that the assumption of shared dynamics does hold.
>
> Additionally, we would not expect small changes in the dynamics across different training samples to have a large impact on the performance of our method. First of all, our approach models the dynamics using neural networks which should be able to handle certain variations in the data. To handle larger variations, we could add more edge-types $e$ to the model. For example, if we were to model the dynamics of how neurons react to the activities of other neurons, it would certainly be helpful to include separate edge types to model excitatory and inhibitory connections between neurons. But we could also add more edge types to additional model connections of differing strengths. Alternatively, if we have prior knowledge about the changes that might happen to the dynamics, we could follow an approach similar to what we present in section 5.2.1. Here, we assume to know that causal relations differ in their strength, and we enable the model to handle this variation through an additional latent variable.
>
> 3 - It is indeed possible to use a recursive decoder that also considers the history of previous values. To keep our setup simple, we decided to use a non-recurrent decoder that makes use of the velocity information instead.
>
> 4 - We agree that it would be very interesting to have theoretical guarantees on recovering the correct causal graph under hidden confounding. In our paper, however, we had a different focus in mind. We wanted to show the promising benefits that Amortized Causal Discovery can bring to the field of causal discovery, and hidden confounders are a long standing challenge in this field. As such, we believe that the presented empirical evidence showing that our model can learn to handle hidden confounders (by leveraging the information that is shared between samples with different underlying causal graphs) is a big step forward in itself.
>
> 5 - We have updated our paper to incorporate the proposed changes.

---

### Official Review · AnonReviewer2 · 2020-10-29
**Causal discovery based on shared dynamics is exciting but does this method work on real(istic) data in the presence of noise?**

**Rating:** 5
**Confidence:** 4

**Review:**

The paper makes an observation that signal dynamics common to a class of causal systems may contain strong information to enable the use of the encoder of the Neural Relational Inference (2018) for extracting (Granger) causal graphs. It minimally extends the NRI model with an empty edge type and demonstrates that the observation holds in a few cases of dynamical systems. Additionally, to handle the unobserved common causes the paper shows improvements in ROCAUC when the encoder is modified to model them directly.

The paper discusses a potentially interesting use of the Neural Relational Inference for (Granger) causal discovery in multivariate time series. Although lacking in detail, it is very clearly written. The proposed method of causal discovery for Markov order one system does demonstrate impressive performance on smooth dynamical systems.

Possibly, the greatest weakness of the NRI approach in this formulation is the seeming reliance on relatively smooth noiseless dynamics. While for the NRI in the original paper (2018) it was within the proposed scope, the claimed use of NRI in this paper is not fully supported by experiments. All of them are from a smooth dynamical system. The Particles experiment is no different from NRI (2018) with the same reported AUC result. Even the most realistic dynamical system from Smith et al. 2011, which is also quite simplistic by the way, still model hemodynamic lag and thus a relatively smooth signal. Notably, this last less smooth of all presented signal is where the NRI model performs the worst.

-   How would NRI in this proposed application behave if we use a still simplistic but potentially less smooth VAR model? Would be good to test it with different SNR.
-   In general, even in dynamical systems used for method demonstration, what happens at various noise levels?
-   How would it behave when the assumption of the model is slightly violated and the signal is generated by an SVAR model?

Given the above limitations, it is unclear if the paper presents something substantially novel/interesting. Technically it is still the NRI model of 2018, so not much contribution there. That would not be a problem if the paper would manage to indeed demonstrate that the model is useful for causal discovery in multivariate time-series. The experiments are limited to very specific kinds of signals and thus not convincing of the approach generality.

It is not immediately clear from preliminaries, whether the proposed model is capable to recover relations of a Markov order higher than one. The described "summary graph" as a compressed representation only holds true to the Markov order one underlying dynamic Bayesian network-like structure (see D. Danks and S. Plis. Learning causal structure from undersampled time series. In NIPS 2013 Workshop on Causality, 2013.), yet the description of the model is a bit vague and may indicate that the method works with the full unrolled graph and then folds it into Markov order one. This part, despite the extensive use of notation, is not clear. It would be simpler to say that indeed an unrolled representation of a graph of limited Markov order is used. It is not clear though if there are any restrictions on the Markov order as discussed since the encoded graph holds just Markov order one.

While the problem of unobserved common causes is discussed, only a single type is considered - fully missing random variables (or particles). Yet, a simple mismatch in measurement and causal time scales will violate some of the assumptions (e.g. generate samples from the VAR model and drop every other time sample for all variables and the model can be fit with SVAR and not VAR). See:

-   M. Gong, K. Zhang, B. Schoelkopf, D. Tao, and P. Geiger. Discovering temporal causal relations from subsampled data. In Proc. ICML, volume 37 of JMLR W&CP, pages 1898–1906. JMLR.org, 2015
-   S. Plis, D. Danks, C. Freeman, and V. Calhoun. Rate-agnostic (causal) structure learning. In Proc. NIPS, pages 3285–3293. Curran Associates, Inc., 2015

From the paper it is unclear what encoder and decoder models were used and can be used to perform the experiments or use the approach. How does the graph enter the decoder together with a variable-length input? If the Decoder is MLP the input layer has a fixed size. Possibly, the MLP is fed one sample at a time with a vector representation of the encoded graph, but that contradicts the paper description. Generally unclear. Supplement mentions that the models are exactly the same as for the NRI paper, but this is still not enough information and does not make the paper self-contained.

From the description of experiments it is unclear how many random variables were used and how many can the NRI handle.

What was the experiment in Figure 3? How many particles/random variables? Was the training done on a single causal generative graph while testing performed on a variety of different unseen graphs? Was it a random sample of a large number of different graphs for both testing and training? Is NRI even capable of learning transferable information from data generated from a single graph?

-   Note, reference P. Spirtes, C. N. Glymour, R. Scheines, and D. Heckerman. Causation, prediction, and search. MIT Press, 2000. - Heckerman is *not* a co-author of [this book](https://mitpress.mit.edu/books/causation-prediction-and-search-second-edition)
-   Last paragraph of page 3 $x\in \mathbb{X}$, the $\mathbb{X}$ is not defined.

---

> ### Author Response · Authors · 2020-11-16
> **Review is copy-pasted, responses to points addressed in previous versions, and more responses**
>
> We thank the reviewer for their feedback, although we note that this review is a near-identical (character-by-character) match to a review we received for a previous submission of this work. We encourage the reviewer to take a closer look at our *current* submission (ICLR 2021), as well as our previous rebuttal, and let us know if any of the raised issues have not been addressed yet.. In particular, there are a number of points which were either addressed in our previous rebuttal (as they were already present in the previous submission), or have been updated in this version:
>
> - The encoder and decoder architectures are discussed in detail in Appendix B.1.2 (without reference to the NRI architecture)
> - The number of random variables used in the experiments are discussed in Appendix B.1.1 - N = 5 for particles/Kuramoto, N = 15 for Netsim
> - The experiment in Fig 3 was done on the particles and Kuramoto datasets (see Fig 3, caption, line 1)
> The review claims that our Particles experiment is “no different from NRI (2018)”. On the contrary, it is different, and, we believe, more difficult: our dataset simulates uni-direction (i.e. causal) relations, instead of bi-directional (i.e. correlational) ones as in NRI (Sec. 5.1 and appendix B.1.1).
> - The review claims that we do not demonstrate causal discovery in multivariate time-series. We note that our Particles experiment is multivariate, with the particles moving around a two-dimensional space (Sec. 5.1, page 7, 2nd paragraph)
> - The reviewer asks if our method is “capable of learning transferable information from a single graph”. We point to the Netsim experiment, which has a single underlying graph (see last paragraph of Sec 5.1), where our method performs nearly as well as the best baseline, despite a small dataset size - we suspect the performance would only improve if more data were available, given the results in Fig. 3.
>
> We additionally respond to several other points in the review:
> - We agree that developing models for noisier settings is a good direction for future work. Nonetheless, causal discovery is still a challenging task in the noiseless setting. We experimentally demonstrate success of our method on a task where many other methods struggle, despite the low-noise setting. On Netsim, the signal to noise ratio is about 100/1: we suspect the poorer performance is due to the smaller dataset size (50 observations with a single underlying causal graph - see bottom of Sec. 5.1) rather than the noisy dynamics.  Therefore, Netsim is not the setting our method is intended to excel on, but it nonetheless performs well.
> - To clarify, the inferred graph does not “enter the decoder” as an input: rather, the sampled graph variables $z_{ij}$ dictate which vertices’ feature vectors will be input into the decoder (see Eq 12)
> - We agree with the reviewer that a mis-match in time scale will violate some of our assumptions. We see this as a good direction for future exploration. We found the task of identifying causal relations with a missing variable challenging and believe that it represents a significant empirical demonstration/contribution on its own.

---

### Author Response · Authors · 2020-11-18
**General comment - our contribution is primarily empirical, and this is consistent with prior work in this line**

We thank all the reviewers for their time and generous feedback. We are glad that the reviewers found the paper clear and the ideas interesting. We would like to respond here to the general comment that our paper does not contain a sufficient theoretical validation of our method, such as in the form of identifiability or consistency proofs.

- We consider our contribution to be empirical - we propose and attack a novel version of the causal discovery problem, where each sample has its own causal graph, and demonstrate that our method outperforms prior methods by a large margin in this setting.

- We compare to a number of prior recent work in the Granger causal discovery literature (e.g. Tank et al, Khanna et al), none of which provide proofs of consistency or identifiability; as such, we believe that our focus on the empirical evaluation of our approach is in line with previous works in this area.

- Additionally, we demonstrate some success in the hidden confounding setting, which (as far as we know), no other prior paper has been able to do.

---

### Decision · Program_Chairs · 2021-01-07
**Final Decision**

**Decision:**

Reject

**Comment:**

The paper proposes an interesting approach that leverages shared dynamics across causal systems for improved joint causal discovery.  The reviewers and AC all agree that the approach is interesting, promising and that the paper is well written.

While theoretical validation would be an exciting thing to have, it is perfectly acceptable for the paper to focus on an empirical study. But in this case, it is very important to provide convincing evaluation experiments.  As several reviewers have pointed out, in order to convincingly demonstrate the value of the approach, it would be very important for the experiments to go beyond noiseless systems.  We strongly encourage the authors to address this point as this will significantly strengthen the significance of their contributions.